# The Influence of Probiotic Lactobacilli on COVID-19 and the Microbiota

**DOI:** 10.3390/nu16091350

**Published:** 2024-04-30

**Authors:** Clarissa Reginato Taufer, Juliana da Silva, Pabulo Henrique Rampelotto

**Affiliations:** 1Graduate Program in Genetics and Molecular Biology, Universidade Federal do Rio Grande do Sul, Porto Alegre 91501-970, Brazil; 2Graduate Program in Health and Human Development, Universidade La Salle, Canoas 92010-000, Brazil; 3Bioinformatics and Biostatistics Core Facility, Instituto de Ciências Básicas da Saúde, Universidade Federal do Rio Grande do Sul, Porto Alegre 91501-970, Brazil

**Keywords:** SARS-CoV-2, bacteria, *Lactobacillus*, probiotics, microbiome

## Abstract

This comprehensive review explores the potential of using lactobacilli as a probiotic in the management of COVID-19. Our findings suggest that lactobacilli show promise in reducing the risk of death, gastrointestinal and overall symptoms, and respiratory failure, as well as in lowering cytokines and inflammatory markers associated with the disease. The molecular mechanisms by which lactobacilli protect against COVID-19 and other viral infections may be related to the reduction in inflammation, modulation of the immune response, and direct interaction with viruses to produce antiviral substances. However, the selected studies demonstrate the presence of mixed findings for various clinical, biochemical, hematological, and immunological parameters, which may be attributed to methodological differences among studies. We highlight the importance of clearly describing randomization processes to minimize bias and caution against small sample sizes and inappropriate statistical tests that could lead to errors. This review offers valuable insights into the therapeutic potential of lactobacilli in the context of COVID-19 and identifies avenues for further research and applications. These findings hold promise for the development of novel approaches to managing COVID-19 and warrant further investigation into the potential benefits of lactobacilli in combating the disease.

## 1. Introduction

The ongoing COVID-19 pandemic has presented a substantial global health challenge, with millions of individuals affected and numerous lives lost. As researchers continue to explore various strategies to combat this viral infection, the potential role of probiotics has garnered considerable attention. Probiotics, which are beneficial microorganisms renowned for their favorable impacts on gut health, are being studied for their immunomodulatory properties and their ability to influence the respiratory and gastrointestinal systems, as well as promising effects in the prevention and management of various infectious diseases [1]. Among the diverse probiotic bacteria, lactobacilli stand out as a prominent candidate due to their well-documented immunomodulatory and anti-inflammatory properties [2].

Recently, there has been a restructuring in the Lactobacillaceae family and the *Lactobacillus* genus to incorporate the genetic diversity revealed by advancements in molecular identification techniques [3]. The term ‘lactobacilli’ is used to collectively refer to all 25 genera within the *Lactobacillus* genus. The recent updates to the taxonomic classification of *Lactobacillus* have not yet been included in the reference databases, resulting in studies continuing to rely on the previous *Lactobacillus* classification. Consequently, in this review, we have opted to maintain the term ‘Lactobacillus’ when referring to the results of previous studies while using ‘lactobacilli’ in the broader context.

Lactobacilli species are host-adapted, especially in vertebrates. They are Gram-positive, homo-fermentative, non-sporulating, and can ferment a diverse array of substrates [3]. They are regularly present in the gastrointestinal tract and other mucosal surfaces of humans, and the decline in lactobacilli in the gut microbiota is commonly associated with various diseases [4]. The diverse species and strains of lactobacilli may influence disease outcomes differently, leading to variations in their impact on health conditions [5]. These bacteria have been extensively studied for their probiotic properties and exert their advantageous effects via multiple processes, including competitive inhibition of pathogenic microorganisms, enhancement of the gut barrier function, and modulation of the host immune response [6].

Given that COVID-19 primarily affects the respiratory system but can also involve gastrointestinal symptoms, the potential role of lactobacilli in mitigating both local and systemic manifestations of the disease is of particular interest. Understanding the specific mechanisms by which probiotic lactobacilli exert its effects on COVID-19 will provide valuable insights into its therapeutic potential. An increasing amount of evidence indicates that the dysregulation of the immune response plays a crucial role in the pathogenesis and severity of COVID-19. Probiotics, including lactobacilli strains, have been shown to regulate immune responses by stimulating innate and adaptive immunity and promoting a balanced inflammatory reaction [7,8]. By boosting the generation of anti-inflammatory cytokines and reducing the release of pro-inflammatory mediators, lactobacilli might be involved in mitigating the hyperactive immune response observed in severe cases of COVID-19. Furthermore, lactobacilli strains have been identified as having the ability to enhance mucosal immunity [9], which could be particularly relevant for preventing viral entry and replication in respiratory and gastrointestinal tissues.

While several studies have explored the potential advantages of probiotics in COVID-19 management [10,11], there remains a need for a compilation and examination of the evidence currently present, specifically focusing on the lactobacilli. The purpose of this review is to connect this gap by critically evaluating the existing literature and to provide a full overview of the beneficial effects of lactobacilli on COVID-19 outcomes, shedding light on its potential as an adjunctive therapeutic option. By consolidating and analyzing the current evidence, this review seeks to contribute to our understanding of probiotic-based interventions in COVID-19 management and guide future research endeavors in this field.

## 2. Search Strategy and Selection Process

We selected the terms “COVID-19” or “SARS-CoV-2” and “Lactobacillus” to conduct a literature search. The strategy construction began with the PubMed database using MeSH terms (Medical Subject Headings) and was adjusted for other databases. In addition to the PubMed database, the search was conducted using the Embase, Web of Science, and Scopus databases, with advanced search settings when possible. Only articles released between January 2020 and November 2023 were included in the search parameters. The initial search was conducted without language limitations, but only studies written in English were evaluated. Appendix A contains the search strategy in detail. The studies selected for review included those that utilized lactobacilli either alone or in combination with other genera for the probiotic treatment of COVID-19 patients at any severity level. Notes, letters, editorials, animal studies, book chapters, review articles, in vitro studies, and case reports were also excluded. Additionally, articles cited by the articles returned in the searches were evaluated and selected if they met the criteria for inclusion established a priori.

Initially, we used the Rayyan software to assess all the articles returned to select and exclude duplicates [12]. Subsequently, the articles were evaluated by two reviewers (C.R.T. and P.H.R.) based on their abstracts or full text, if necessary. Articles that met the inclusion criteria underwent data extraction based on the full text.

For each article, the data were synthesized in an Excel spreadsheet, including details regarding the study information, methodology, and results, as shown in Appendix A.

## 3. Selected Studies

Our search returned 475 studies across the four databases. After eliminating duplicates, 319 papers were evaluated. Additionally, four studies were identified via citations. In total, we initially evaluated 323 studies, of which 291 were excluded based on abstract evaluation and did not meet the requirements for inclusion. Following the screening process, eleven papers were included in this review, as illustrated in Figure 1.

The results are presented in two tables. Table 1 presents eleven studies specifically examining the use of probiotic strains of lactobacilli for the management of COVID-19 [13,14,15,16,17,18,19,20,21,22,23]. A single study examined the utilization of three strains of lactobacilli; the remaining studies assessed a combination of probiotics, which included various species of lactobacilli along with another genus. Italy has the highest number of studies among the four [13,14,15,16], followed by Iran [17,18] and China [19,20] (two studies); Belgium [21], Spain [22], and Russia [23] presented just one. There were six prospective studies [16,17,18,21,22,23], four retrospective studies [13,14,19,20], and one post-COVID-19 study [15].

With the exception of Navarro-López et al. (2022), who confirmed COVID-19 patients by antigen test diagnosis or polymerase chain reaction, all other studies were confirmed using RT-PCR for SARS-CoV-2 [22]. In only two cases, patients used probiotics at home, as observed in the study by De Boeck et al. (2022), which assessed the use of throat spray probiotics, where the patients themselves collected and stored the samples at home during the study, and in the study by Laterza et al. (2023), in which the evaluation consisted of patients who had been discharged from the hospital after being hospitalized for COVID-19 [15,21]. The remaining studies were conducted with hospitalized patients [13,14,16,17,18,19,20].

Only three studies reported the severity of COVID-19 cases, with one evaluating the use of probiotics in mild patients, one in severe patients, and one in severe to critical cases [16,19,20]. The time of probiotic treatment varied from 10 days to up to 30 days [16,22]. Ivanskin et al. (2021) considered the treatment endpoint on the 14th day of hospitalization or on the day the patient was discharged or died; however, the group had a mean length of hospital stay of 11 days (10–14 days) [23]. Two studies did not clarify the treatment duration [13,20].

The selected studies employed a variety of statistical methods to analyze the impact of probiotic lactobacilli on COVID-19 and the microbiota. Strengths of the statistical approaches include the utilization of advanced techniques such as Benjamini–Hochberg FDR correction, General Linear Mixed models, Mann–Whitney U-test, Cox regression models, *t*-tests, and Chi-square tests. These methods allowed for the robust analysis of various outcomes such as mortality, inflammatory markers, symptom improvement, and microbial composition. However, some limitations should be considered. The studies may have had small sample sizes, potentially restricting the applicability of the findings. Additionally, the use of multiple statistical tests without appropriate adjustments for multiple comparisons could increase the risk of type I errors. For these reasons, while the statistical methods employed offered valuable perceptions into the role of probiotic lactobacilli on COVID-19, careful consideration of their strengths and limitations is essential for interpreting the results accurately.

Session 4 will discuss the findings in detail. Table 2 summarizes the results of the clinical, biochemical, and hematological/immunological parameters evaluated in each study for the administration of probiotics.

## 4. The Probiotic Lactobacilli in the Management of COVID-19

The ongoing COVID-19 pandemic has led to a critical need for efficient therapeutic strategies to manage the disease. Probiotics, particularly lactobacilli species, have garnered attention for their potential immunomodulatory and antiviral properties, raising the question of whether they could be involved in COVID-19 management. In this section, we discussed the findings of the selected studies that investigated the potential role of probiotic lactobacilli strains in the management of COVID-19. We first examined the results of each study and then provided a general critical discussion about the main outcomes.

### 4.1. Study Results Analysis

d’Etorre et al. (2020) assessed the use of the commercial probiotic Sivomixx^®^, which contains five strains of lactobacilli, two strains of *Bifidobacterium*, and *Streptococcus thermophilus* [14]. The use of the probiotic led to the cessation of diarrhea in all patients within 7 days. More than 40% (6/14) resolved within 24 h after the start of treatment, and almost all (13/14) within 3 days. Other symptoms (such as asthenia, headache, fever, dyspnea, and myalgia) showed similar trends from the 2nd day of treatment. Regarding the respiratory outcome (evaluated with the General Linear Mixed model using the GLIMMIX procedure), there was a significant difference in the evolution between the groups (*p* < 0.001). The risk of the condition progressing to respiratory failure was significantly reduced by 8 times after 7 days of treatment. Although not statistically significant, the group that did not receive probiotic treatment had a higher number of patients who required being moved to the intensive care unit (ICU) for mechanical ventilation (2/42, 4.8%) or experienced a fatal outcome (4/42, 9.5%). No patient who received probiotic treatment required ICU admission or died.

The same commercial probiotic was evaluated by Cecarelli et al. (2021) in a retrospective study regarding the in-hospital mortality outcome in COVID-19 patients [13]. The groups were divided into patients with or without the administration of the commercial probiotic Sivomixx^®^, in addition to the administration of the best available therapy (BAT). In total, 44 patients died (22%); among these, 34 (30%) were patients who did not receive bacteriotherapy, and only 10 (11%) were treated with oral bacteriotherapy (*p* < 0.001). The reduction in the risk of death was also confirmed after adjustments for age, Charlson, Call, PSI, CURB, and CURB-65 scores. However, there was no differential risk regarding the need for ICU admission between patients in the probiotic and non-probiotic groups. Additionally, regarding the incidence of fungal and bacterial superinfections in the ICU, there were no significant differences between the groups. Furthermore, the length of hospitalization was longer for patients who received BAT and oral bacteriotherapy (20 days) compared to patients who received only BAT (14 days) (*p* < 0.001).

Regarding the hospitalization period and duration of symptoms, the evaluation of the probiotic LactoCare^®^, containing a mixture of seven strains of lactobacilli, four strains of *Bifidobacterium*, and *S. thermophilus*, did not show significant differences [18]. However, the assessment of the proinflammatory cytokine interleukin-6 (IL-6) was significantly lower in the symbiotic group than in the placebo group (*p* = 0.045). Additionally, a significant difference in the reduction in IL-6 levels during the evaluation period was observed for the probiotic group (*p* = 0.005), which did not occur for the non-probiotic group. Also, the IL-6 changes were statistically significant between the groups (*p* = 0.002). One patient from each group required ICU admission, and a patient from the placebo group remained hospitalized for 14 days. There were no between-group differences in the assessment of symptoms at 0 and 14 days for both the probiotic and non-probiotic groups. Within the probiotic group, there was a significant decrease in white blood cell counts during the intervention (*p* = 0.004), which was not observed in the control group, but no significant differences were observed between the groups. For the other parameters evaluated (alanine aminotransferase, creatinine, hemoglobin, aspartate aminotransferase, platelet count, blood urea nitrogen, erythrocyte sedimentation rate, C-reactive protein, polymorphonuclear neutrophil, and lymphocytes), there were no significant differences between the groups at the end of the treatment.

The patients’ progress was assessed via the combined intake of the yeast *Kluyveromyces marxianus* B0399 and *Lactobacillus rhamnosus* CECT 30579 by Navarro-López et al. (2022) [22]. The authors evaluated the resolution of both digestive and non-digestive symptoms in COVID-19 patients. Concerning gastrointestinal symptoms, significant differences in the improvement in heartburn and abdominal pain symptoms were observed in comparisons between the groups. In the non-probiotic group, 33.3% (1/3) of patients showed improvement in heartburn symptoms in the second assessment, compared to 90% (9/10) of patients with improvement in symptoms in the treated group. Similarly, 62.5% (5/8) of placebo patients showed improvement in abdominal pain compared to 100% (9) of patients with improvement in the probiotic group. However, there was no statistical difference when assessing the total number of patients with any intestinal symptoms and improvement in digestive symptoms between the groups. Regarding evaluations of non-digestive symptoms (such as fever, anosmia/ageusia, respiratory symptoms, chest pain, conjunctivitis, and other pains), 92.8% (13/15) of placebo group patients still had some symptoms, compared to 58.3% (14/24) of patients in the probiotic group (*p* = 0.06). In the assessment of the overall symptom evolution, there was a statistical difference between the groups, with an improvement of 70.82% vs. 88.55% in the probiotic and placebo groups, respectively. No ICU admissions or deaths were recorded in this study.

De Boeck et al. (2022) used the administration of *Lactobacillus casei* AMBR2, *L. rhamnosus* GG, and *Lactobacillus plantarum* WCFS1 in throat spray format to evaluate the severity of symptoms [21]. The same trends were observed in both groups without significant differences, as well as the time for improvement. The use of the spray was also assessed for test positivity and its relation to other symptoms. After one week of the study, 73% of the probiotic group and 77% of the placebo group tested positive (*p* = 1). After two weeks of the study, 17% and 32% of the participants in the probiotic and placebo groups, respectively, still had positive results for the SAR-CoV-2 test (*p* = 0.22). At the end of the study, 6.7% of patients in the probiotic group (2/30) and 26% (7/27) in the placebo group still had positive tests for SAR-CoV-2 (*p* = 0.07). All symptoms showed a strong correlation with positivity, independent of the intervention. Nose/throat microbial composition did not show significant variations in composition during viral infection or microbiome treatment by Principal Coordinates Analysis (PCoA). Assessing the abundance of amplicon sequence variants (ASVs) of the strains administered in this study at different times, significant variations were noted between the groups. The mean relative abundances for the *L. casei* ASV was 1.6%, *L. plantarum* ASV was 1.3%, and *L. rhamnosus* ASV was 0.5% in the probiotic group throughout the entire study. In the placebo group, these values were below 0.01% for all three ASVs. Prevalence analysis based on sequencing data was 38.6% for *L. casei* ASV1, 28% for *L. plantarum* ASV2, and 13.4% for *L. rhamnosus* ASV4 for the probiotic group, while for the placebo group, these values were 10.5%, 7%, and 2%. The association of the relative abundances of a selection of important airway ASVs (*Haemophilus*, *Dolosigranulum*, *Rothia*, *Streptococcus*, *Moraxella*, *Staphylococcus*, and lactobacilli) with severity scores, treatment, and viral loads were also assessed during all study periods. ASVs that showed the largest effect size in the probiotic group were *Moraxella* ASV4 (*M. lacunata*), *Rothia* ASV14 (*R. amarae*), and various commensal *Streptococcus* ASVs (*S. thermophilus*, *S. rubneri*, and *S. sanguinis*, among others), in addition to deliberately added lactobacilli *L. casei* ASV, *L. plantarum* ASV2, and *L. rhamnosus* ASV4. Significant negative associations with treatment were also identified, with stronger effects observed for *Dolosigranulum* ASV1 (*D. pigrum*), *Streptococcus* ASV6 (*S. crispatus*, *S. oligofermentans*, and *S. sinensis*), and *Streptococcus* ASV7 (*S. gordonii*). Associations for symptom scores and specific taxa showed significant negative results (*p* < 0.05) with moderate effect sizes. A significant negative association was identified between ASVs corresponding to the administered lactobacilli and the acute symptom score, suggesting that these lactobacilli may cause symptoms to become less severe. After 3 weeks, there were no notable differences in the IgG antibody response between the groups.

The assessment of symptoms, renal function, liver function, and inflammatory markers was the main outcome evaluated during a study of a probiotic mix containing *L. rhamnosus* and *Bifidobacterium* [23]. It was found that the duration of viral diarrhea was reduced by an average of two days and hospital-acquired diarrhea was prevented for patients taking a single antibiotic. However, there was no significant effect on the course of the disease and inflammatory biomarkers. The probiotic also did not have a significant effect on survival rates and kidney and liver dysfunction.

Also, for the evaluation of symptoms and inflammatory markers, Saviano et al. (2022) assessed the use of the probiotic called Lactibiane Iki^®^, which contains *Lactobacillus salivarius* LA 302, *Lactobacillus acidophilus* LA 201, and *Bifidobacterium lactis* LA 304, for 10 days [16]. The non-probiotic group showed a significantly higher increase of 199% (*p* = 0.005) in fecal calprotectin compared to only a 29% increase in the probiotic group on days 3–5. From days 3–5 to days 7–10, the reduction in the level of the inflammatory marker was 35% in the probiotic group and only 16% in the non-probiotic group (*p* = 0.006). Between the groups, there was also a similar level of C-reactive protein (CRP) at the recruitment time in the probiotic group versus the non-probiotic group. At days 3–5, the mean level decreased by 72.7% in the probiotic group compared to 62% in the non-probiotic group (*p* < 0.001). In evaluations between days 7–10, the mean level was 5 ± 3 mg/dl in the probiotic group compared to 9 ± 2 mg/dl in the non-probiotic group (*p* < 0.05). Despite the abnormal white blood cell count in both groups, no significant changes were observed between the groups during the evaluation period. Moreover, in the group that received the probiotic, a faster and continuous reduction in the need for oxygen support was observed, while in the non-probiotic group, the decrease was smaller. In this context, four patients in the probiotic group (10%) between days three and six of the evaluation needed oxygen support at a level of 60% with a high-flow nasal cannula and/or positive pressure mechanical breathing, and three patients (7.5%) remained in the ICU for at least 24 h. One patient (1.25%) died after a follow-up period of 60 days. In the probiotic group, no patient required oxygen support at 60% or more. Regarding the average length of hospitalization, it was 14 ± 6 days for the probiotic group, while the non-probiotic group was 19.0 ± 10 days (*p* = 0.52).

Inflammatory markers and white blood cells were also assessed after 8 weeks of supplementation with a mix of Lactobacillus bulgaricus, L. casei, L. acidophilus, L. rhamnosus, S. thermophilus, Bifidobacterium breve, and B. longum [17]. There was a decrease in the levels of C-reactive protein (from 6.9 ± 20.6 to 3.3 ± 1.8 mg/L), erythrocyte sedimentation rate (Δ = 6.2 ± 17.0 mm/h), and IL-6 (Δ = 0.6 ± 10.4 pg/mL), but the results were not statistically significant (*p* > 0.05). The white blood cell count increased at the end of the treatment (an increase of 915 ± 2462 cells/L), but it was also not statistically significant (*p* > 0.05).

The use of probiotics was also specifically analyzed in patients with severe COVID-19. The recommendation to use probiotics came from the Chinese management guidelines for COVID-19 [24]. The treatment and duration were determined by the attending physician. The probiotics included combined oral tablets of lactobacilli, *Bifidobacterium*, *Enterococcus*, and *Bacillus* [17,19]. Dynamic changes in eight hematological/immunological parameters (B lymphocytes, NK cells, total T lymphocytes, CD4+ T cells, CD8+ T cells, CD4/CD8 ratio, IL-6, and CRP) were evaluated at three different time points: upon hospital admission (T1), midway through the entire hospitalization (T2), and the last test before hospital discharge (T3). The IL-6 value gradually increased over time for the treatment group. On the other hand, NK cells, T lymphocytes, and B lymphocytes were upregulated from T1 to T2 in the probiotic-treated group but did not differ from the results at T3 compared to the non-probiotic group. CD4+ T cells were upregulated from T2 in the probiotic group, and CD8+ T cells were upregulated in the non-probiotic group, but they did not differ from T3 values. The CD4+/CD8+ ratio stayed within the normal range in probiotic-treated patients, whereas it exceeded the normal range in patients without probiotics.

For a group of severe and critical patients, the use of *B. longum*, *L. bulgaricus*, and *S. thermophilus* was evaluated regarding nutritional status, frequency of diarrhea, inflammatory markers, and other hematological/immunological parameters [20]. Out of the 58 patients analyzed, a total of 16 presented symptoms of diarrhea, with 10 of these patients receiving probiotics while 6 were in the control group. The probiotic group experienced a significantly shorter duration of diarrhea compared to the control group (*p* < 0.05). Inflammatory markers procalcitonin and C-reactive protein were significantly lower in the probiotic group at the end of the treatment (*p* < 0.05), and plasma albumin levels and lymphocyte count were significantly higher in patients treated with probiotics (*p* < 0.05). The duration until the test turned negative for SARS-CoV-2 was also evaluated, and it was significantly shorter in the treated group (*p* < 0.05). No differences were observed between the groups in white blood cells and neutrophil counts.

In post-COVID-19 patients, the commercial probiotic VSL#3^®^ was evaluated [15]. The probiotic mixture contained four strains of lactobacilli spp., three strains of bifidobacteria, and one strain of *S. thermophilus* BT01. At the end of the 8-week treatment, the levels of IL-6, tumor necrosis factor-alpha (TNF-α), and interleukin-12RA (IL-12RA) were significantly reduced. The level of citrulline was also significantly reduced, possibly related to changes in gut microbiota composition and bacterial metabolism. No significant changes were observed in bowel habits and intestinal symptoms.

### 4.2. Critical Discussion on Main Findings

Altogether, these studies indicate that using lactobacilli as a probiotic for managing COVID-19 has shown promise in decreasing mortality risk and gastrointestinal and general symptoms, minimizing the likelihood of respiratory failure, and lowering cytokines and inflammatory markers. These studies have explored various aspects, such as symptom management, disease severity, biochemical parameters, and clinical outcomes.

Nevertheless, when individually analyzed, these different aspects have presented mixed findings. For instance, studies like d’Etorre et al. (2020) and Cecarelli et al. (2021) have shown that probiotic interventions, including Sivomixx^®^, can lead to the resolution of symptoms like diarrhea, asthenia, headache, fever, dyspnea, and myalgia in COVID-19 patients [13,14]. These studies have also suggested that probiotics may help reduce the severity of COVID-19, with lower rates of ICU admissions and in-hospital mortality among patients who received probiotic interventions. However, other studies, like De Boeck et al. (2022), did not find significant differences in disease severity between probiotic and placebo groups [21]. In addition, while some studies have reported improvements in clinical symptoms with probiotic treatment, others have not found significant differences in symptom severity or disease progression between probiotic and control groups [17,19]. Of the 11 studies, 9 demonstrated beneficial effects on the clinical outcomes [13,14,15,16,18,20,21,22,23].

In terms of biochemical and hematological/immunological parameters, studies evaluating LactoCare^®^ and Lactibiane Iki^®^ have shown the potential to decrease inflammatory markers in COVID-19 patients, indicating a potential role of probiotics in modulating the immune response and inflammation [16,18]. On the contrary, studies by Ivashkin et al. (2021), Li et al. (2021), and Vaezi et al. (2023) did not observe any changes in the parameters analyzed [17,18,23]. Of eight studies analyzing biochemical parameters, five did not show significant differences [17,18,19,21,23], while three presented a general reduction or increase in the parameters [15,16,20]. Key biomarkers analyzed in different studies, like IL-6 and C-reactive protein, have mixed results. Of eight studies analyzing hematological/immunological parameters, six did not show significant differences [16,17,18,19,20,23], while one presented a reduction in white blood cells [18] and one an increase in lymphocytes [20].

The heterogeneity of results observed may be attributed to several factors. One significant factor contributing to this variation is the diversity in probiotic strains used across the studies. Various lactobacilli strains may possess distinct immunomodulatory and antiviral properties, leading to differences in their effectiveness in managing COVID-19 symptoms.

Additionally, the duration of the probiotic regimen administered in these studies varied significantly. The optimal duration for probiotic treatment to exert its full therapeutic effects in COVID-19 management remains unclear. Variations in treatment duration among studies can impact the outcomes observed, as the time frame for observing improvements in symptoms or clinical outcomes may differ.

Moreover, the severity of the disease among participants in these studies may also influence the results obtained. Patients with varying degrees of COVID-19 severity may respond differently to probiotic interventions. The effectiveness of probiotics in managing symptoms and disease progression may be more pronounced in milder cases compared to severe cases where multiple factors contribute to the clinical outcomes.

The implications of the heterogeneity of results and methods in these studies are significant. It highlights the need for standardized protocols and guidelines for probiotic interventions in COVID-19 management to ensure consistency and comparability across studies. Future research should focus on identifying the most effective probiotic strains, optimal treatment durations, and patient populations that would benefit the most from probiotic supplementation. By addressing these factors, we can better understand the role of probiotics in managing COVID-19 and enhance the overall efficacy of probiotic interventions in combating the disease.

## 5. Risk of Bias and Quality Assessment

To assess the quality of the selected studies, we utilized Cochrane’s Risk of Bias 2 (RoB2) [25] for randomized intervention studies and ROBINS-I [26] for non-randomized interventional studies. The results are presented in Figure 2.

Ceccarelli et al. (2020) and d’Ettorre et al. (2020) conducted a retrospective assessment of the impacts associated with the utilization of probiotics as a supplementary component in the treatment of COVID-19 [13,14]. Li et al. (2021) also retrospectively evaluated the effects of using probiotics administered to patients under the guidance of the medical team following the Chinese management guideline for COVID-19 (version 7.0) [19].

Ivashkin (2021) conducted a randomized, controlled, single-center, open-label trial, and Navarro-López (2022) conducted a randomized, prospective, open-label, non-blinded study in which patients voluntarily accepted the invitation to participate [22,23]. Both patients and carers and people delivering were aware of the treatment.

Laterza (2023) conducted a non-randomized intervention study that assessed the use of probiotics in patients after hospital discharge (before and 8 weeks after) [15]. The study clearly outlined the inclusion and exclusion criteria and control samples were obtained from the same patients. This approach helped exclude confounding factors, thereby reducing the potential risk of bias.

The most significant restriction found via the risk of bias analysis was related to bias arising from the randomization process. This suggests that the randomization process in the studies may not have been effectively implemented, potentially leading to an unequal distribution of participant characteristics between the treatment and control groups. This type of bias can significantly impact the internal validity of the studies, as it may introduce systematic differences between the groups that could influence the observed outcomes. It underscores the importance of rigorous randomization procedures in ensuring the comparability of treatment groups and minimizing the risk of bias in clinical research.

## 6. The Probiotic Lactobacilli in the Management of Other Viral Infections

The probiotic lactobacilli have garnered attention for their possible contribution to mitigating other viral infections, extending beyond the realm of COVID-19. A growing body of evidence suggests lactobacilli may be essential in controlling the immune system and affecting the course of viral infections.

### 6.1. Clinical Studies

In healthy children (3–6 years), the consumption of fermented dairy beverages *L. casei* reduced the incidence rate of common infectious diseases by 19%. In particular, the treated group exhibited a 24% reduction in the incidence rate of gastrointestinal tract infections and demonstrated an 18% lower incidence rate of upper respiratory tract infections than the control group. Lastly, the active group experienced a 2% decrease in the incidence rate of lower respiratory tract infections in comparison to the control group [27]. The administration of *Lactobacillus* GG as a probiotic was also associated with a significantly reduced risk of acute infectious diarrhea in infants and children [28].

In healthy adults, the consumption of *L. plantarum* HEAL 9 (DSM 15312) and *Lactobacillus paracasei* 8700:2 (DSM 13434) shows a significant reduction in the number of days with common cold symptoms in the incidence of acquiring common cold episodes and the reduction in pharyngeal symptoms. In the probiotic group, the proliferation of B lymphocytes was significantly counteracted in comparison with the control group [29]. The administration of *L. paracasei* ssp. *paracasei* (*L. casei* 431^®^) probiotics also showed an improvement in the immune response after seasonal influenza vaccination in healthy adults [30]. The group treated with probiotics for six weeks exhibited significantly higher levels of vaccine-specific plasma IgG3, IgG1, and IgG.

In elite athletes, the oral administration of *Lactobacillus fermentum* VRI-003 was linked to a substantial decrease in the duration and intensity of respiratory diseases [31].

Among healthy elderly individuals, the consumption of yogurt fermented by *Lactobacillus delbrueckii* ssp. *bulgaricus* OLL1073R-1 revealed a significant decrease in the risk of contracting the common cold in the treated group when compared to those who exclusively consumed milk [32].

### 6.2. Preclinical Studies

Regarding experimental studies, the protective action was also investigated with the administration of different species of lactobacilli. In an animal model, treatment with heat-killed *L. casei* DK128 was shown to protect against H3N2 virus infections. The viral titers were 18 times lower than those in untreated mice, with significantly reduced levels of IL-6 in bronchoalveolar lavage in mice pre-treated with heat-killed *L. casei* BK128. Furthermore, these mice exhibited significantly lower levels of TNF-α compared to untreated mice [33].

For the Influenza virus, the oral use of heat-killed *L. plantarum* L-137 showed viral titers in the lungs were significantly lower compared to controls in the early phase after influenza virus infection and was observed IFN-β in the serum of treated mice, which did not occur in untreated mice [34].

*L. delbrueckii* ssp. *bulgaricus* OLL1073R-1 and its exopolysaccharides orally administering yogurt fermented also exerted an anti-influenza virus effect in mice. Four days post-infection, a notable decrease in virus titer and a significant elevation in anti-influenza virus antibodies were observed in bronchoalveolar lavage fluid. Both groups exhibited increased activity of natural killer (NK) cells from splenocytes. Furthermore, mice treated experienced an extended survival period [35].

In a mouse model exposed to lethal levels of the influenza virus, the oral administration of *Lactobacillus pentosus* b240 significantly prolonged the survival period of the animals. In cases of non-lethal infections, the daily oral intake of *L. pentosus* b240 exhibited a noteworthy, dose-dependent reduction in infectious influenza virus titers in the lungs at 7 days post-infection [36].

Survival protection was evident in models receiving treatment with *L. plantarum* DK119, with a 50% survival rate observed in the group administered a low dose of *L. plantarum* DK119. Mice that underwent intranasal and oral pre-treatment before viral infection demonstrated enhanced survival against lethal influenza virus infection. Moreover, in both treatment approaches, the viral lung load was significantly diminished, ranging from approximately 200 to 800 times lower than untreated counterparts [8].

Routes of administration and the use of live and dead lactobacilli are also important variables in the application of probiotics. Mice subjected to intranasal administration exhibited higher survival rates compared to those receiving oral administration. Additionally, survival rates were consistently higher in mice receiving live lactobacilli compared to those receiving the inactive form, regardless of the route of administration. Specifically, in the oral route, the survival rate was 40% for live lactobacilli and 0% for the inactive form, while in the intranasal route, it was 70% for live lactobacilli and 40% for the inactive form [37].

Also, intranasal inoculations of live *Lactobacillus reuteri* and *L. plantarum* exhibited protective effects against pneumovirus. The pre-treatment of mice revealed a transient recruitment of granulocytes. Additionally, sustained protection against the lethal consequences of the infection was observed, persisting for up to 91 days post-virus inoculation, with a 60% survival rate. Notably, heat-inactivated *L. plantarum* also conferred protection against the lethal virus challenge [38].

Collectively, these findings from clinical and experimental studies indicate that different strains of lactobacilli hold potential advantages in preventing and mitigating viral infections, especially within the gastrointestinal and respiratory tracts. As research progresses, the role of lactobacilli in combating viral infections continues to unfold, presenting a promising avenue for preventive and therapeutic strategies.

## 7. Protective Mechanisms of Probiotic Lactobacilli against COVID-19

It is unclear how exactly lactobacilli protect against COVID-19 and other viral infections, but several hypotheses have been proposed. In this section, we discuss the mechanisms by which lactobacilli may modulate the immune response, reduce inflammation, and exert direct antiviral effects.

### 7.1. Immunomodulatory Mechanisms

Several research studies have investigated the impacts of lactobacilli strains on the immune response and their ability to influence cytokine production [39]. For example, lactobacilli can stimulate the activity of NK cells, which are essential for the early prevention of viral infections [40]. This suggests a specific mechanism involving the development of NK cell activity in response to probiotic components [41].

Different lactobacilli strains exert their immunomodulatory influence via Toll-like receptor 2 (TLR2) binding, recognizing peptidoglycan within Gram-positive bacterial cell walls. Notably, in vitro investigations revealed heightened TLR2 expression in Caco-2 cells (human cell line) following exposure to *L. plantarum* and *L. rhamnosus* [42]. In addition, various lactobacilli strains engage with Toll-like receptor 4 (TLR4) to modulate immune responses.

In vivo, *L. casei* exhibited analogous effects in both *Salmonella*-infected and healthy mice, stimulating TLR expression. For instance, assessment of mice pre- and post-*Salmonella* challenges, *L. casei* exhibited enhanced interleukin-10 (IL-10), IL-6, and interferon-gamma (IFN-γ) production while reducing TNF-α levels via TLR4 interaction [43]. Additionally, *L. rhamnosus* GG (heat-inactivated) and *L. delbrueckii* downregulate TLR4 expression in human monocyte-derived dendritic cells (DCs) [44].

Many lactobacilli species also modulate immune responses via NOD-like receptors. Within swine’s galactose-1-phosphate uridylyl transferase (GALT), *Lactobacillus gasseri* and *L. delbrueckii* upregulate NLRP3 expression via TLR and the NOD signaling cascade, ensuring proper NLRP3 activation [45]. Additionally, agonists of NOD1, NOD2, TLR2, and TLR9 contribute to enhanced NLRP3 expression. Notably, *L. salivarius* promotes the synthesis of IL-10 and regulates NOD2 to have an anti-inflammatory action [46].

Furthermore, by modifying both mucosal and systemic immune responses, lactic acid bacteria, including lactobacilli, show promise for enhancing human and animal health [47]. It can also suppress viral proliferation (Th1 and protective immune responses are triggered by *L. plantarum* strain YU from fermented food) and has shown capabilities to protect cell monolayers against viral rupture, rotavirus, and transmissible gastroenteritis virus [48]. This broader modulation may contribute to enhanced defense against infections, including respiratory viruses like SARS-CoV-2.

Studies focused on the gut–lung axis have also emphasized the capacity of lactobacilli to interact with immune cells and modulate cytokine production, implying a possible part in influencing the immune system’s reaction to diseases like COVID-19 [49]. Lactobacilli may contribute to immune modulation by producing metabolites and short-chain fatty acids that can have immunomodulatory potential [50].

### 7.2. Anti-inflammatory Mechanisms

Lactobacilli exhibits a capacity to enhance anti-inflammatory cytokines while suppressing pro-inflammatory ones. Different strains of lactobacilli have demonstrated modulation of cytokine production. For example, *L. casei* elevates antiviral cytokines, including TNF-α, IL-10, and IFN-γ in healthy mice, while decreasing TNF-α and increasing IFN-γ, IL-6, and IL-10 in *Salmonella*-infected mice [43]. *Lactobacillus rhamnosus* also demonstrates the ability to induce the expression of TNF-α, IL-6, and IL-10 in macrophages and suppresses the expression of TNF-α and IL-6 in LPS-stimulated macrophages [51]. While *L. plantarum* induces the production of interleukin-12 (IL-12) and IFN-γ, it concurrently reduces inflammatory cytokines such as interleukin-4 (IL-4), IL-6, and TNF-α, as identified in bronchoalveolar lavage [8].

Both alive and dead *L. rhamnosus* GG mitigate TNF-α–induced interleukin-8 production [52]. It has also been demonstrated that mycelium fermentation of the *L. rhamnosus* EH8 strain produces butyric acid, which can down-regulate IL-6 secretion in macrophages and the expression of PDE4B [53].

Regulation of the nuclear factor kappa-light-chain (NF-κB) pathway is a key mechanism in this context. The NF-κB pathway plays a crucial role in numerous pathological conditions, governing the expression of around 150 anti-inflammatory and pro-inflammatory cytokine genes; numerous lactobacilli strains actively modulate NF-κB pathway activation. *L. casei* demonstrates inhibition of *Shigella flexneri* induced NF-κB pathway activation [54]. *L. rhamnosus* and *Lactobacillus helveticus* can decrease the Th1 pro-inflammatory response while enhancing the Th2 response in the event of a *Citrobacter rodentium* infection [55]. *L. casei*, *L. reuteri*, and *L. paracasei* exhibit anti-inflammatory properties via NF-κB pathway modulation. For example, *L. reuteri* reduces the level of inflammatory mRNA cytokines, enhances anti-inflammatory cytokine synthesis, and boosts apoptosis-inhibiting proteins, promoting cell survival and immune response. Also, it achieves this by disrupting IκB ubiquitination and p65 nuclear translocation (NF-κB subunit) [56,57]. In parallel, *L. casei* and *L. paracasei* prevent pro-inflammatory cytokine production by inhibiting IκBα phosphorylation and p65 nuclear translocation, also reversing IκBα degradation [58,59]. Similar NF-κB pathway inhibitory effects are observed with *L. plantarum* and *Lactobacillus brevis*. By reducing NF-κB binding activity, *L. plantarum* lowers NF-κB-activating factors [60], while *L. brevis* prevents interleukin 1 receptor-associated kinase 1 (IRAK1) and AKT phosphorylation [61].

### 7.3. Direct Antiviral Mechanisms

Understanding the probiotic lactobacilli’s direct antiviral mechanisms to COVID-19 involves the production of antiviral substances, competition with pathogenic viruses for cellular binding sites, and interference with viral replication. For example, certain lactobacilli species can inhibit viral entry and replication by producing antimicrobial peptides and other substances [6].

Additionally, the ability of lactobacilli to compete with pathogenic viruses for binding sites on host cells may limit viral attachment and entry. It has been demonstrated that certain lactobacilli strains generate substances such as bacteriocins and exopolysaccharides that can inhibit viral replication and protect against infection [62]. Also, some strains of lactobacilli can bind to specific receptors on the enterocyte and stimulate MUC-2 and MUC-3 genes, resulting in increased inhibition of bacterial translocation and inhibition of enteropathogen adhesion to intestinal epithelial cells [63,64].

In addition, the release of ACE-inhibiting peptides by lactobacilli may represent one of the potential ways in which probiotics work to prevent the entry of the virus into cells. For example, it has been reported that numerous probiotics, particularly lactic acid bacteria, can generate peptides that exhibit ACE inhibitory effects [65]. Furthermore, a recent computational study using multiple mechanistic approaches to probiotic metabolites has provided additional support for this concept. Anwar et al. (2021) conducted a molecular docking analysis of the antiviral effect of metabolites in *L. plantarum*, proving how well they work to prevent the spread of viruses into cells by binding to ACE2, RdRp, and RBD [66].

Another suggested mechanism against SAR-CoV-2 is via the direct interference of probiotics in viral replication, as exemplified by Lactococcin G in *L. plantarum*. This antimicrobial peptide exhibits high affinities for binding to viral proteins. The study by Balmeh et al. (2021) not only identifies Lactococcin G’s potential but also highlights the expanded use of probiotic bacteria’s modified bio-antimicrobial peptides as potential COVID-19 medications [67]. Moreover, the antiviral activity of *L. plantarum* Probio-88, also explored via molecular docking, further demonstrates the possible therapeutic benefit of adjuvant probiotics in viral infections [68].

## 8. NGS Approaches to Unraveling the Effects of Probiotic Lactobacilli on COVID-19

As previously discussed, the specific mechanisms through which probiotic lactobacilli exert their effects on COVID-19 pathophysiology infections are not fully understood, indicating a significant research gap in the field. To address this gap, the use of NGS technologies emerges as a cutting-edge strategy [69]. Integration of NGS methodologies into the exploration of probiotic lactobacilli’s impact on COVID-19 pathophysiology presents a groundbreaking approach to unraveling the complex interplay between probiotics and viral infections. By exploring the microbial environment at a genetic level, NGS provides a comprehensive view of the dynamic changes occurring within the microbiota in response to probiotic interventions [70].

In addition, future studies could focus on elucidating the genomic, transcriptomic, and epigenomic markers linked to COVID-19, offering further insights into the underlying pathophysiological mechanisms of the disease. By investigating the genetic, gene expression, and epigenetic profiles associated with COVID-19, it is possible to uncover essential information regarding the molecular signatures and regulatory processes involved in the progression and outcomes of the infection. Such comprehensive analyses have the potential to unveil novel biomarkers, therapeutic targets, and pathways that could enhance our understanding of COVID-19 pathophysiology and pave the way for more effective diagnostic and treatment strategies [71,72].

## 9. Conclusions

In summary, the research found in this review suggests that the use of lactobacilli as a probiotic in the management of COVID-19 has demonstrated its potential to reduce the risk of death, gastrointestinal and overall symptoms, lower the risk of developing respiratory failure and reduce cytokines and inflammatory markers. However, these studies have presented mixed findings for clinical, biochemical, hematological, and immunological parameters, which can be attributed to the variety of methodologies applied among them. Additionally, the randomization processes must be clearly described to minimize the risk of bias. Furthermore, the small sample sizes of some studies and the use of multiple statistical tests without appropriate adjustments for multiple comparisons could increase the risk of type I errors. Lactobacilli have the potential to provide protection against COVID-19 via their capability to reduce inflammation, modulate the immune response, and directly interact with viruses to produce antiviral substances. Via this analysis, we offer an exploration of the therapeutic potential of lactobacilli in the context of COVID-19, providing valuable insights and shedding light on potential directions for further study and use, as well as possible avenues for future research and applications.

## Figures and Tables

**Figure 1 nutrients-16-01350-f001:**
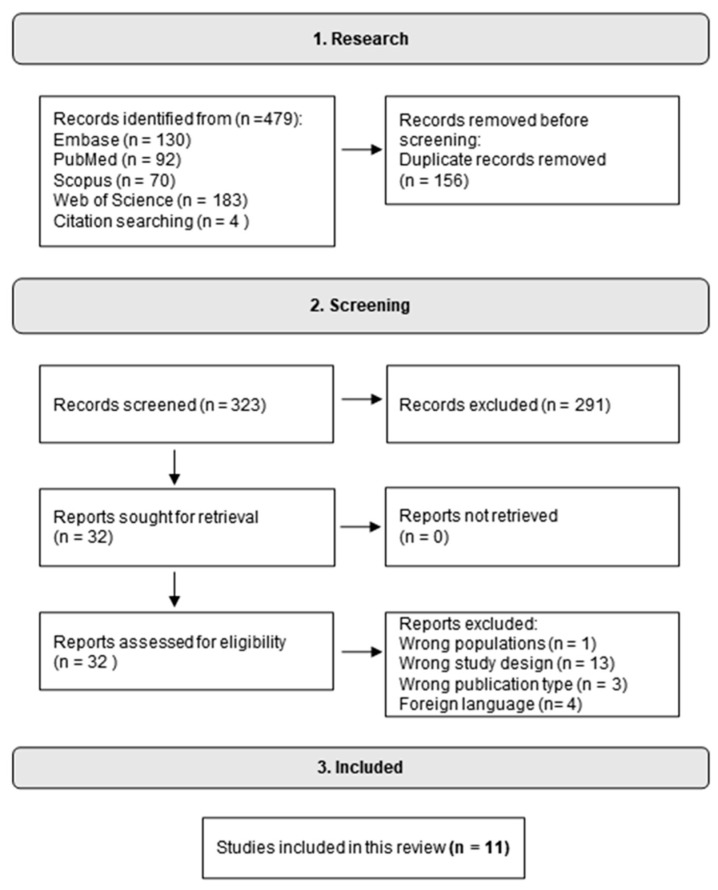
Flow diagram of the database search protocol.

**Figure 2 nutrients-16-01350-f002:**
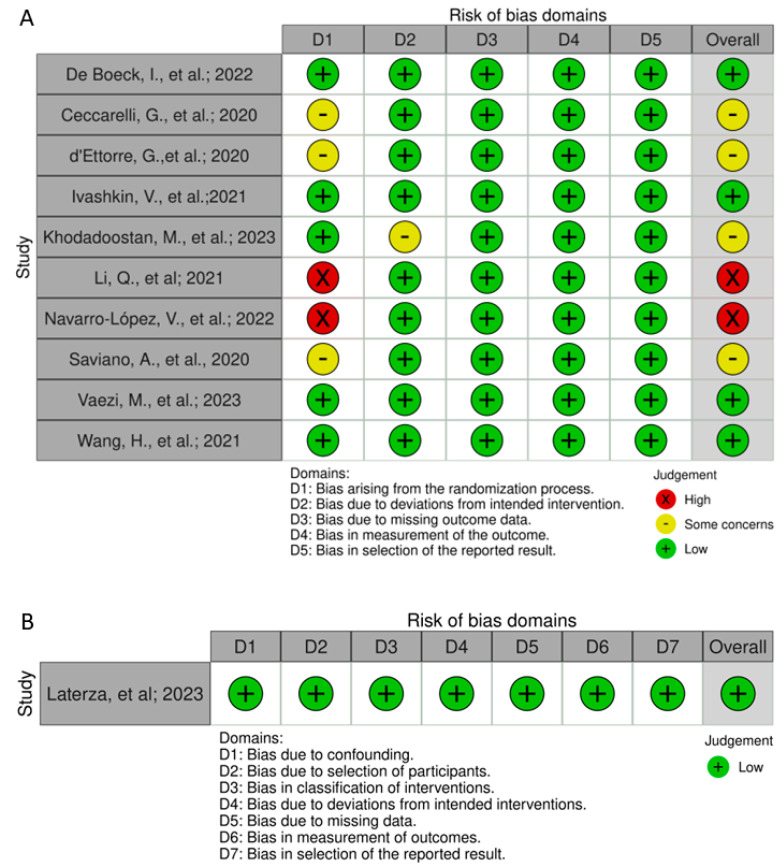
Traffic lights chart for risk of bias for randomized intervention studies [13,14,16,17,18,19,20,21,22,23] (**A**) and non-randomized intervention studies [15] (**B**).

**Table 1 nutrients-16-01350-t001:** Information extracted from the literature regarding article specifics and corresponding research findings for lactobacilli as probiotic strains.

Study	Country	N ^1^	COVID-19 Severity	Groups	Probiotics	Probiotic Regimen Duration	Outcome
Ceccarelli, G. et al., 2020 [13]	Italy	200	Missing information	88 probiotic group 112 non-probiotic group	*Lactobacillus acidophilus* DSM 32241, *Lactobacillus brevis* DSM 27961, *Bifidobacterium lactis* DSM 32246 and DSM 32247, *Lactobacillus helveticus* DSM 32242, *Lactobacillus paracasei* DSM 32243, *Lactobacillus plantarum* DSM 32244, *Streptococcus thermophilus* DSM 32245, (Sivomixx^®^, Ormendes SA, Jouxtens-Mézery, Switzerland)	Not informed	Significant reduction in the risk of death
De Boeck, I. et al., 2022 [21]	Belgium	64	Missing information	34 probiotic group 30 non-probiotic group	*Lacticaseibacillus casei* AMBR2, *Lacticaseibacillus rhamnosus* GG, and *Lactiplantibacillus plantarum* WCFS1	14 days	Score of acute symptoms negatively associated with administered lactobacilli
d’Ettorre, G. et al., 2020 [14]	Italy	70	Missing information	28 probiotic group 42 non-probiotic group	*Lactobacillus acidophilus* DSM 32241, *Lactobacillus brevis* DSM 27961, *Bifidobacterium lactis* DSM 32246 and DSM 32247, *Lactobacillus helveticus* DSM 32242, *Lactobacillus paracasei* DSM 32243, *Lactobacillus plantarum* DSM 32244, *Streptococcus thermophilus* DSM 32245, (Sivomixx^®^, Ormendes SA, Jouxtens-Mézery, Switzerland)	14 days	Remission of diarrhea, reduction in other symptoms, 8x lower risk of developing respiratory failure
Ivashkin, V. et al., 2021 [23]	Russian	200	Missing information	99 probiotic group 101 non-probiotic group	*Lacticaseibacillus rhamnosus* PDV 1705, *Bifidobacterium longum* subsp. *longum* PDV 2301, *Bifidobacterium bifidum* PDV 0903, and *Bifidobacterium longum* subsp. *infantis* PDV 1911 (Florasan-D)	No more than 14 days	Average reduction of two days in the duration of viral diarrhea and prevention of hospital-acquired diarrhea for patients receiving a single antibiotic
Khodadoostan, M. et al., 2023 [17]	Iran	55	Missing information	28 probiotic group 27 non-probiotic group	*Lactobacillus casei*, *Lactobacillus acidophilus*, *Lactobacillus bulgaricus*, *Lactobacillus rhamnosus*, *Bifidobacterium longum*, *Bifidobacterium breve*, and *Streptococcus thermophilus*	8 weeks	No significant differences were observed
Laterza, L. et al., 2023 [15]	Italy	19	Missing information	19 post-COVID-19 patients	*Lactobacillus paracasei* BP07, *Lactobacillus helveticus* BD08, *Lactobacillus plantarum* BP06, *Lactobacillus acidophilus* BA05, *Bifidobacterium animalis* subsp. *lactis* BL03, *Bifidobacterium breve* BB02, *Bifidobacterium animalis* subsp. *lactis* BI04, and *Streptococcus thermophilus* BT01 (VSL#3^®^, lot number 909031, VSL Pharmaceuticals, Gaithersburg, MD, USA)	8 weeks	Significant reduction in citrulline, TNF-ALFA, IL-6, and IL-12RA
Li, Q. et al., 2021 [19]	China	311	Severe	123 probiotic group 188 non-probiotic group	*Lactobacillus acidophilus*, *Lactobacillus bulgaricus*, *Bacillus cereus*, *Bacillus subtilis Bifidobacterium infantis*, *Dung enterococcus*, *Bifidobacterium longum*, *Streptococcus termophiles*, and *Enterococcus faecium*	Mean duration was 12.94 days	No significant differences were observed
Navarro-López, V. et al., 2022 [22]	Spain	41	Missing information	26 (24 ^1^) probiotics group 15 non-probiotic group	*Lactobacillus rhamnosus* CECT 30579 and *Kluyveromyces marxianus* B0399	30 days	Improvement in symptoms of pyrosis and abdominal pain; overall symptom improvement
Saviano, A. et al., 2022 [16]	Italy	80	Mild	40 probiotic group 40 non-probiotic group	*Lactobacillus salivarius* LA 302, *Lactobacillus acidophilus* LA 201, and *Bifidobacterium lactis* LA 304 (Lactibiane Iki^®^, PiLeJe, Champtoceaux, France)	10 days	Lower mean length of hospitalization, faster and continuous reduction needed for O_2_ support, reduction in the inflammatory marker CRP, lower values of fecal calprotectin
Vaezi, M. et al., 2023 [18]	Iran	76	Missing information	38 probiotic group 38 non-probiotic group	*Lactobacillus rhamnosus*, *Lactobacillu helveticus*, *Lactobacillus casei*, *Bifidobacterium lactis*, *Lactobacillus acidophilus*, *Bifidobacterium breve*, *Lactobacillus bulgaricus*, *Bifidobacteriumlongum*, *Lactobacillus plantarum*, *Bifidobacterium bifidum*, *Lactobacillus gasseri*, and *Streptococcus* *thermophilus* (LactoCare^®^, Kindstrom-Schmoll, Sycamore, IL, USA)	14 days	Reduction in IL-6 levels and white blood cell count
Wang, H. et al., 2021 [20]	China	58	Severe to critical	23 probiotic group 35 non-probiotic group	*Bifidobacterium longum*, live *Lactobacillus bulgaricus*, and *Streptococcus thermophilus*	Not informed	Reduction in diarrhea and inflammatory markers PCT and CRP; increase in albumin levels and lymphocyte count. Shorter time to negative SARS-CoV-2 test

^1^ N: patients analyzed.

**Table 2 nutrients-16-01350-t002:** Clinical, biochemical, and hematological/immunological parameters analyzed in each selected study. Green circles with upward arrows indicate an increase in the measured parameter, red circles with downward arrows indicate a reduction, and gray circles with a horizontal line indicate no change. ICU: intensive care unit; IgG: Immunoglobulin G; IL-6: Interleukin-6; IL-12RA: Interleukin-12RA; TNF-α: Tumor Necrosis Factor-alpha.

Study	Clinical Parameters	Biochemical Parameters	Hematological and Immunological Parameters
Ceccarelli, G., et al., 2020 [13]	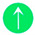 Length of hospital stays		
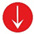 Risk of death		
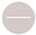 Incidence of ICU, incidence of bacterial and fungal superinfections in the ICU		
De Boeck, I., et al., 2022 [21]	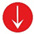 Overall symptoms	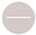 IgG antibody response	
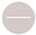 Severity of symptoms, time for improvement		
d’Ettorre, G., et al., 2020 [14]	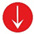 Overall symptoms, diarrhea, respiratory outcome		
Ivashkin, V., et al., 2021 [23]	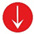 Hospital-acquired diarrhea (received only one antibiotic), diarrhea	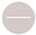 C-reactive protein, creatinine, ferritin, fibrinogen, alanine aminotransferase, albumin, aspartate aminotransferase, total bilirubin	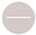 Erythrocyte sedimentation rate, white blood cells, neutrophils, lymphocytes
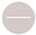 Duration of illness, length of hospital stays, incidence of ICU admission, oxygen support or need for mechanical ventilation, survival rates		
Khodadoostan, M., et al., 2023 [17]		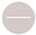 IL-6, C-reactive protein	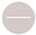 Erythrocyte sedimentation rate, white blood cells
Laterza, L., et al., 2023 [15]	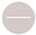 Gastrointestinal symptoms	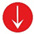 IL-6, TNF-α, IL-12RA, citrulline	
Li, Q., et al., 2021 [19]		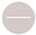 IL-6	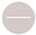 Lymphocytes, natural killer cells, CD4+ T cells, CD8+ T cells, CD4+/CD8+ ratio
Navarro-López, V., et al., 2022 [22]	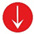 Overall symptoms, overall symptom evolution, gastrointestinal symptoms		
Saviano, A., et al., 2022 [16]	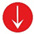 Oxygen support, length of hospital stays	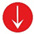 C-reactive protein, calprotectin	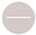 White blood cells
Vaezi, M., et al., 2023 [18]	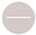 Overall symptoms, duration of symptoms, length of hospital stays	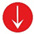 IL-6	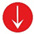 White blood cells
	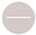 C-reactive protein, alanine aminotransferase, aspartate aminotransferase, creatinine	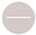 Hemoglobin, lymphocytes, platelet, erythrocyte sedimentation rate, polymorphonuclear neutrophil, blood urea nitrogen
Wang, H., et al., 2021 [20]	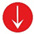 Diarrhea, time to negative SARS-CoV-2 test	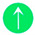 Albumin	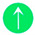 Lymphocytes count
	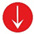 C -reactive protein, procalcitonin	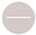 White blood cells count and neutrophil count

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
