# Peer review of "The Influence of Probiotic Lactobacilli on COVID-19 and the Microbiota"

_nutrients, 2024, doi:10.3390/nu16091350_

Round 1
Reviewer 1 Report
Comments and Suggestions for Authors
Dear Redactors,
Thank you very much for the opportunity to revise the article „The Impact of Probiotic Lactobacilli on COVID-19 and the Microbiota”. Taufer et al. aimed to investigate the potential of lactobacilli species as a probiotic for managing COVID-19.
The article is very interesting and well written. I have just a few comments.
In the introduction describe Lactobacilli in more details.
Table 2 is hard to read and illegable. Please, make it more readible.
I get the feeling that role of Lactobacilli species in inflammation was not describe enough. Please, add some more details abou this.
Thanks.
Author Response
Dear Redactors,
Thank you very much for the opportunity to revise the article „The Impact of Probiotic Lactobacilli on COVID-19 and the Microbiota”. Taufer et al. aimed to investigate the potential of lactobacilli species as a probiotic for managing COVID-19.
The article is very interesting and well written. I have just a few comments.
Reply: We thank the reviewer for the comments and suggestions. We have considered them all and have revised the manuscript accordingly. Below we provide the point-by-point reply to all comments and suggestions. Changes in the revised version of the manuscript are highlighted in red.
In the introduction describe Lactobacilli in more details.
Reply: We included some additional information l.47-53.
Table 2 is hard to read and illegable. Please, make it more readible.
Reply: We changed the table to make it more readable.
I get the feeling that role of Lactobacilli species in inflammation was not describe enough. Please, add some more details about this.
Reply: We included additional information l.569-576.
Reviewer 2 Report
Comments and Suggestions for Authors
*This study has an important topic that will be of interest to readers.
*Standard headers for sections would improve readability.
*It appears that you conducted a systematic review. It is recommended that you add that to the title and use PRISMA guidelines and checklist for reporting. If you did not conduct a systematic review indicate the type of review you conducted. I was expecting a standard format for a systematic review.
*Abstract
*Seems overly abbreviated and is missing needed content. Structured headers to your abstract would aid the reader. More detail about the studies reviewed is needed. What types of studies and how many participants etc. What are the main findings?
* "Lactobacilli associated with other species" needs clarity. This reviewer took that to mean that Lactobacillus was given in combination with other probiotics. It looks like Lacto probiotics were given in combination in all of the included in all but one study where three strains of Facto were given.
*Introduction. The following statement is unclear- "The recent changes in the taxonomic 40 classification of Lactobacillus mean that the reference databases still were not updated, and 41 studies still use the Lactobacillus classification." Perhaps this reclassification of Lactobacilli should be removed if it is not relevant?
*Method/Search strategy. You show a PRISMA Diagram therefore it looks like you have done a systematic review.
*Table 1 is mostly well done. "Not informed" is unclear in this context. Do you mean the information was missing from the articles? If so generally we use "not applicable or "missing" of. Please indicated the type of study in a column (RCT vs cohort etc). In the last column on clinical relevance "beneficial" is a judgement that you should allow the reader to make. It is better to remain objective in the Table 1 findings. In your discussion you could indicate how many of the studies has findings of significant improvement in COVID symptoms/duration etc.
Table 2 was very difficult for this reviewer to see and interpret. It appears there is significant heterogeneity in these outcomes. The symbols were not explained in a key and it was difficult to discern meaning from this Table in the present format. It is suggested that you find a way to present this so that the reader can see patterns. If you symbols are indicating findings then be sure to describe that while introducing the Table/Figure. Is there a way to include this in Table 1 so that all data is in one place?
Is Section 4 this statement is unclear. "In this section, 160 we explored the results of the selected studies that investigated the potential role of 161 probiotic lactobacilli strains in the management of COVID-19." Section 4 appears to be detailed findings of each of the included studies. This section would be more interesting, useful and readable if there was less redundancy with Tables 1 and 2 and the findings were more integrated. For example, rather than describing each study in detail, organize the paragraphs by topics (eg. COVID severity, COVID duration, clinical parameters biochemical parameters etc. You could also include a paragraph on benefits of the intervention.
*Risk of Bias. The information below the table about the types of studies should move up so the reader can evaluate the scores for risk of bias knowing the type of study. This reviewer questions the accuracy of the risk of bias ratings. It seems unlikely that so many of these studies would have such low risk of bias since only 2 were RCTs and none were double-blind placebo controlled trials.
Section 6 and 7 contains interesting background information on Lactobacilli against viruses.
Section 8 does not bring in new information that has not been discussed in the article.
Section 9. The content of the conclusions are good, but seems that the information in the conclusions was not clearly described in the paper. As it reads now, it seems like the conclusions go beyond the review presented. This may be because the authors failed to clearly present these main areas in body of the review. This reviewer recommends that the authors could use the these points in the conclusions to enhance the content of the other sections. For example, potential to reduce death an morbidities was not clearly presented previously. The impact on laboratory measures could also have been more clearly presented in the body of the review. The limitation presented in the conclusions were not discussed earlier.
Author Response
*This study has an important topic that will be of interest to readers.
Reply: We thank the reviewer for the comments and suggestions. We have considered them all and have revised the manuscript accordingly. Below we provide the point-by-point reply to all comments and suggestions. Changes in the revised version of the manuscript are highlighted in red.
*Standard headers for sections would improve readability.
Reply: This is a comprehensive review and not a systematic review. That’s why it does not have the standard format for a systematic review.
*It appears that you conducted a systematic review. It is recommended that you add that to the title and use PRISMA guidelines and checklist for reporting. If you did not conduct a systematic review indicate the type of review you conducted. I was expecting a standard format for a systematic review.
Reply: This is a comprehensive review and not a systematic review. That’s why it does not have the standard format for a systematic review.
*Abstract
*Seems overly abbreviated and is missing needed content. Structured headers to your abstract would aid the reader. More detail about the studies reviewed is needed. What types of studies and how many participants etc. What are the main findings?
Reply: We have rewritten the abstract to include the missing information.
* "Lactobacilli associated with other species" needs clarity. This reviewer took that to mean that Lactobacillus was given in combination with other probiotics. It looks like Lacto probiotics were given in combination in all of the included in all but one study where three strains of Lacto were given.
Reply: We changed this sentence.
*Introduction. The following statement is unclear- "The recent changes in the taxonomic classification of Lactobacillus mean that the reference databases still were not updated, and studies still use the Lactobacillus classification." Perhaps this reclassification of Lactobacilli should be removed if it is not relevant?
Reply: We would rather keep this information, so we have rewritten it to make it clearer. L.42-44.
*Method/Search strategy. You show a PRISMA Diagram therefore it looks like you have done a systematic review.
Reply: We performed a structured systematic search only in part of the review, i.e. related to the use of Lacto as a probiotic in COVID-19. The part about other viral infections and the molecular mechanism was a comprehensive review. That’s why we can’t consider the whole manuscript as a systematic review.
*Table 1 is mostly well done. "Not informed" is unclear in this context. Do you mean the information was missing from the articles? If so generally we use "not applicable or "missing" of. Please indicated the type of study in a column (RCT vs cohort etc). In the last column on clinical relevance "beneficial" is a judgement that you should allow the reader to make. It is better to remain objective in the Table 1 findings. In your discussion you could indicate how many of the studies has findings of significant improvement in COVID symptoms/duration etc.
Reply: All suggestions were addressed. We included the discussion info in a new section 4.2. l.349-403.
Table 2 was very difficult for this reviewer to see and interpret. It appears there is significant heterogeneity in these outcomes. The symbols were not explained in a key and it was difficult to discern meaning from this Table in the present format. It is suggested that you find a way to present this so that the reader can see patterns. If you symbols are indicating findings then be sure to describe that while introducing the Table/Figure. Is there a way to include this in Table 1 so that all data is in one place?
Reply: We changed the table to make it more readable and included an explanation for the symbols. We would rather keep both tables as separate files.
Is Section 4 this statement is unclear. "In this section, we explored the results of the selected studies that investigated the potential role of probiotic lactobacilli strains in the management of COVID-19." Section 4 appears to be detailed findings of each of the included studies. This section would be more interesting, useful and readable if there was less redundancy with Tables 1 and 2 and the findings were more integrated. For example, rather than describing each study in detail, organize the paragraphs by topics (eg. COVID severity, COVID duration, clinical parameters biochemical parameters etc. You could also include a paragraph on benefits of the intervention.
Reply: We have better written that sentence and have structured the section in two parts. In 4.1, we kept the detailed findings of each of the included studies as we consider them relevant. However, we included a second part (4.2) collectively discussing the findings and the implications of the heterogeneity found in results and methods.
*Risk of Bias. The information below the table about the types of studies should move up so the reader can evaluate the scores for risk of bias knowing the type of study. This reviewer questions the accuracy of the risk of bias ratings. It seems unlikely that so many of these studies would have such low risk of bias since only 2 were RCTs and none were double-blind placebo controlled trials.
Reply: The assessment of bias risk for each evaluated item was conducted through a set of questions, and the result was automatically generated by the assessment tool. Upon reviewing the questions and answers, we observed that even if some of the questions are responded to as "not informed" or "probably," the result generated by the tool remains ‘low’.
Section 6 and 7 contains interesting background information on Lactobacilli against viruses. Section 8 does not bring in new information that has not been discussed in the article.
Reply: We would rather keep section 8 to better highlight the relevance of NGS approaches to unraveling the effects of probiotic lactobacilli on COVID-19.
Section 9. The content of the conclusions are good, but seems that the information in the conclusions was not clearly described in the paper. As it reads now, it seems like the conclusions go beyond the review presented. This may be because the authors failed to clearly present these main areas in body of the review. This reviewer recommends that the authors could use the these points in the conclusions to enhance the content of the other sections. For example, potential to reduce death an morbidities was not clearly presented previously. The impact on laboratory measures could also have been more clearly presented in the body of the review. The limitation presented in the conclusions were not discussed earlier.
Reply: We changed the abstract and included additional information in the discussion (including a new section 4.2) to better reflect the conclusions.
Round 2
Reviewer 2 Report
Comments and Suggestions for Authors
The edits significantly enhanced the paper. Very interesting and readable. The integrated findings will be very useful to researchers.